

# Comparing the runoff decompositions of small testbed catchments: end-member mixing analysis against hydrological modelling

Andrey Bugaets[1], Boris Gartsman[1,2], Tatiana Gubareva[1,2], Sergei Lupakov[1], Andrey Kalugin[2], Vladimir Shamov[1], Leonid Gonchukov[1,2]

[1]Pacific Institute of Geography, FEB RAS, Vladivostok, 690041, Russia
[2] Water Problems Institute, RAS, Moscow, 117971, Russia

*Correspondence to*: Andrey Bugaets (andreybugaets@yandex.ru)

**Abstract**. This study is focused on the comparison of catchment streamflow composition simulated with three well-known rainfall-runoff (RR) models (ECOMAG, HBV, SWAT) against hydrograph decomposition onto the principal constituents evaluated from End-Member Mixing Analysis (EMMA). There used the data provided by the short-term in-situ observations at two small mountain-taiga experimental catchments located in the south of Pacific Russia. All used RR models demonstrate that two neighboring small catchments disagree significantly in mutual dynamics of the runoff fractions due to geological and landscape structure differences. The geochemical analysis confirmed the differences in runoff generation processes at both studied catchments. The assessment of proximity of the runoff constituents to the hydrograph decomposition with the EMMA that makes a basis for the RR models benchmark analysis. We applied three data aggregation intervals (season, month and pentad) to find a reasonable data generalization period ensuring results clarity. In terms of runoff composition, the most conformable RR model to EMMA is found to be ECOMAG, HBV gets close to reflect specific runoff events well enough, SWAT gives distinctive behavior against other models. The study shows that along with using the standard efficiency criteria reflected proximity of simulated and modelling values of runoff, compliance with the EMMA results might give useful auxiliary information for hydrological modelling results validation.

## 1 Introduction

Understanding the runoff composition is a fundamental problem in hydrological science (Heidbüchel et al., 2012). Most of the processes in hydrological systems takes place underground and only a limited range of measurement techniques are available; the observation data need to be extrapolating in both space and time to predict the response of any catchment to a given rainfall event with adequate uncertainty (Perrin et al., 2003). Contribution of different water masses to total runoff is time-dependent owing to the temporal variability of flow response processes. In spite of many publications, it is still challenge task to determine how a catchment generates runoff as a whole system. There is an undeniable consensus in the hydrologic community that the experimental catchments research plays a key role for the understanding of hydrological processes and provides essential outdoor laboratories for validation flow pathways and residence times (Gartsman, Shamov, 2015; Holzmann, 2018; Széles et al., 2020).



Sophisticated hydrological models are powerful tools to test hypotheses about catchment function and dealing with uncertainties throughout the observation–conceptualization–modeling sequence (Dunn et al., 2008). Plausibly simulated streamflow can be resulted in different runoff compositions from models with distinct structures, that is widely known as equifinality. To narrowing down the equifinality, it is necessary to evaluate the simulation results - how realistically it reflects

the natural behavior of a hydrological system. However, the selection of the hydrological models and the model structure uncertainties are still not fully understood (Johnson et al., 2003; Butts et al., 2004; Clark et al., 2008, 2011; Li et al., 2015). Aggregation of outcome from different models or evolving model structure along axis of complexity using "top-down'' strategy may lead to parsimonious model that provides useful insights into a catchment behavior (Atkinson et al., 2002; Sivapalan et al., 2003; Marshall et al., 2006; Bai et al., 2009; Clark et al., 2011).

Catchment hydrology is still very much empirical science (Hornberger et al., 1985), and it is common that some parts of a conceptual model may be more rigorously based on physical theory than others. Empirical approaches are still the base or part of the numerous well-known rainfall–runoff (RR) models. The main reason to hang on this empiric knowledge is the scale-dependency of HRU-based model, where they more appropriate than any small-scale physical laws (Beven, 2012; Beven and Germann, 2013). Model parameters are seldom directly measurable and are inferred from calibration. They are generally

model-specific and represent an average behavior in terms of both spatial and temporal variability (Vrugt et al., 2005). Model structure uncertainty is as important as the parameter uncertainty, if mathematical formulation is not based on fundamental laws but adopt a set of lumped functions relating impact to response (Beven, 2012).

All these pitfalls led to a desire in alternative means of model evaluation in terms of how well a model captures the partitioning, storage and release of water by a catchment (Lischeid, 2008). End-member mixing analysis can provide estimates

of the relative contributions of direct (surface) flow, deep ground flow and soil percolation flow to a total catchment runoff (Clark and Fritz, 1997; Evans and Davies, 1998; Burns et al., 2001; Soulsby et al., 2011). Numerous researchers indicated the importance of geochemistry for the transit time estimation and recognized the utility of tracers as additional independent measure for the model evaluation (McGuire et al., 2007; Fenicia et al., 2008; Fenicia et al., 2010; McDonnell et al., 2010; Soulsby et al., 2011; McMillan et al., 2012). There are a number of examples of using tracers to augment modeling but their

evident advance restrained by the lack of suitable datasets. Gradual improvements in the reliability and economics of field and laboratory methods provide now the necessary fine-time resolution data series (Birkel and Soulsby, 2015; Beven, 2019).

This study is focus on comparison the catchment streamflow composition simulated with three well-known RR models against the hydrograph decomposition obtained from End-Member Mixing Analysis (EMMA) to rank hydrological models according to the real processes of runoff generation. In general, it relates to the problem of harmonizing various approaches

based on the solution of direct or inverse tasks of modeling. In the ideal case, the results of solving the direct and inverse tasks should be the same (closely match). The paper is laid out as follows: next section describes the case study catchments and field observation details; all used models (EMMA, ECOMAG, SWAT, and HBV) are outlined in Section 3; EMMA model results are presented in Section 4 while the simulation results of RR models - in Section 5 (for the clarity sake, the results of





hydrological models calibration are placed in the appendix); all results are generalized and discussed in Section 6; and finally,

a summary and some concluding remarks are provided.

## 2 Case study: objects, data and tools description

Studied territory relates to the Pacific Russia boreal forests that is influenced by the East Asian Monsoon. There for two small river catchments (Elovy (3.5 km²) and Medvezhy (7.6 km²) creeks) belonging to the Upper-Ussuri Biocenological Experimental Station (45 km², 44°02' N, 134°11' E) runoff modeling and hydrograph separation were performed. Considered

area is characterized by mid-mountainous relief with moderately steep (locally very steep) hillslopes (Kozhevnikova, 2009; Gartsman and Shamov, 2015). The average altitudes are 500–700 m a.s.l, maximal values reach 1100 m a.s.l.

Air temperature goes through strong variability from year to year; average, absolute minimal and maximal values are +0.7°C, -45 °C (in January) +38 °C (July-August). The annual average precipitation amount is 700-800 mm up to 80% of which occurs in warm period (from late April to October) in liquid form. Unstable intra-annual and long-term precipitation

regimes of territory define runoff formation conditions of investigated catchments. Resources of groundwater are not significant due to fracture rock. Small rivers runoff can be vanishingly small during winter or summer drought period and reach 30-50 mm/day during high intensity rainfall caused by tropical cyclones – typhoons –activity. The range of maximal daily heavy rains is 100–200 mm. The stable snowpack usually occurs in November, snow cover depth reaches up 0.5–1 m by the end of March, common range of snow water equivalent is 100-200 mm. The average values of soil freezing are 1.0–1.2 m.

Field observations were carried out from March to October of 2012-2018 years. Two weather stations were mounted at ~650 (Elovy catchment) and 750 m a.s.l. (Medvezhy catchment) to record air temperature and humidity, wind characteristics and precipitation volume. Streams outlets were fitted with hydrostatic water level loggers. Observations time resolution was 15 min. Measurements of water discharges were run once in 2–4 days during low water period, daily during average moisture conditions, twice a day for flood period. To obtain daily discharges for whole observation period $Q=f(H)$ relationship was

applied. Rain, stream and soil water were sampled while discharges measuring.

Hydrological properties of each soil horizons were obtained based on fair profiles number (Bugaets et al., 2021) and digital soil map was developed (Fig. 1). The soil cover of considered territory is dominated by the Cambisol group (IUSS Working Group WRB, 2015). Brown forest soils here are characterized by high rock fragments content (greater than 80% of profiles volume) which leads to quick infiltration rates. The parent rocks (eluvium and eluvium-deluvium) are well water

permeable due to high porosity.

Soil water samples were taken with vacuum lysimeters installed at the depth of 0.35–0.80 m. Streamflow temperature, pH, specific conductance and TDS were measured in situ. Hydrocarbonate was determined from unfiltered probes by potentiometric titration in the wake of sampling. Samples of ions $HCO_3^-$, $NO_3^-$, $Cl^-$, $SO_4^{2-}$, $Ca^2$, $Mg^{2+}$, $K^+$, $Na^+$ , content of dissolved Si, and dissolved organic carbon (DOC) were collected using 0.45 µm filters and analyzed in the lab (Boldeskul et

al., 2014; Gubareva et al., 2015, 2016).





Data quality control suggested next simulation periods: from 2012 to 2014 for Elovy creek and from 2015 to 2017 for Medvezhy creek. WMO 31939 weather station (namely Chuguevka) data was used to fill up observation gaps and cold part of the years. WRF-ARW model output was applied to derive daily solar radiation interpolated from closest computational grid nodes.

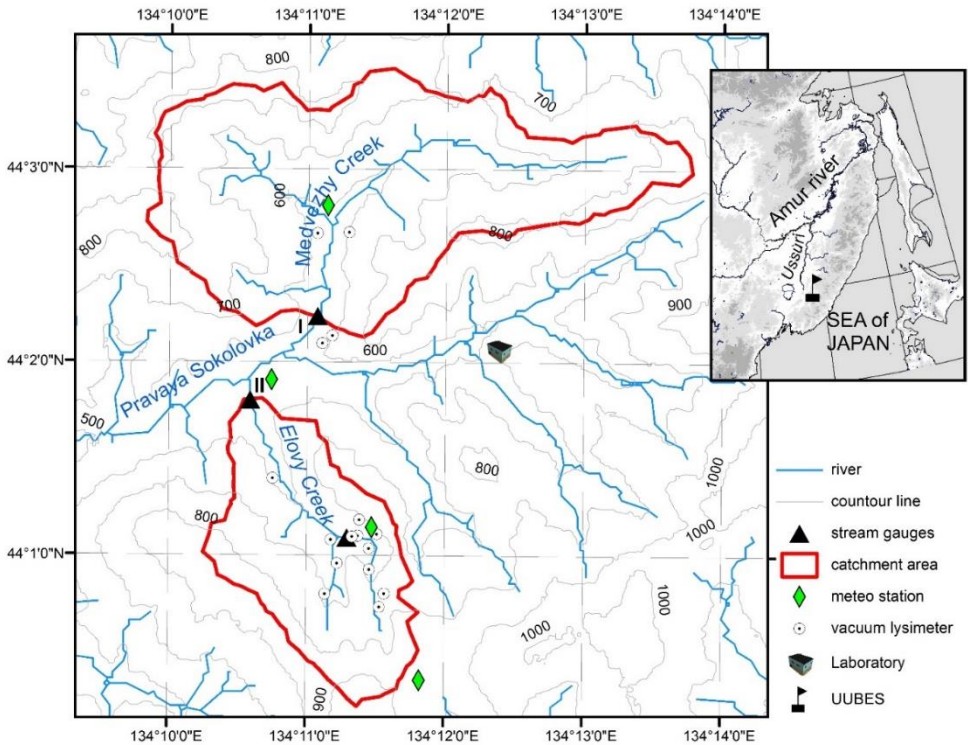


**Figure 1: Schematic map of Pravaya Sokolovka river at the Upper-Ussuri Biocenological Experimental Station, Russian Far East, showing location of sampling and monitoring sites. Catchments: I – Medvezhy, II Elovy.**

## 3. Methods of modelling

In this study, three RR models (SWAT, HBV and ECOMAG) represent a wide range of conceptual diversity of approaches are used to solving the direct task of runoff modeling. HBV is lumped storage based, and other two are considered as spatial distributed (HRU based) models. They differ in structure and methods of runoff generation mechanisms parameterization, physical and mathematical foundation.

The end-member mixing analysis (EMMA) framework is an example of solving the inverse task of the modeling with using tracer hydrology methods. In opposite to the imitation RR models, EMMA can be called an identification model, providing the interpretation and quantification of runoff generation sources (hereafter called fractions for simplicity). It is substantial, the discharges of different runoff sources are determined from direct observations of the water and tracers volumes



with using the mass balance equations system and some empirical relations. Overall, this could be interpreted as sophisticated way of the runoff sources observation.

All listed models are well known in the hydrological community and were previously applied in the region (Gubareva et al., 2015, 2016; Lupakov et al., 2021; Bugaets et al., 2018, 2019, 2021). For description of the models structure, variables and parameters the original terms from the base publications are used. Detailed information can be obtained by references.

### 3.1. EMMA model

End-Member Mixing Analysis (EMMA) is a commonly applied method to identify and quantify the dominant runoff generation sources. It is based on assumption that waters, flowing through specific compartments of watershed, acquire its unique chemical signatures and these can be used as tracers to determine the contributions of each source (labeled as the «end-member») into the total runoff. This technique of hydrograph separation is based on the mass balance approach, which assumes: 1) tracers behave conservatively, i.e. geochemical signature of water contributing from various sources is constant and unique to make it distinguishable from each other; 2) mass conservation law applies both to the water quantity and water quality including mixing of water masses without their transformation (Christophersen et al, 1990; Christophersen and Hooper, 1992).

Hooper (2003) suggested that assumptions of linearity of mixing and conservative behavior of tracers can be evaluated using bivariate scatter plots and residuals derived from the selected model. Bivariate scatter plots should be developed for all potential combination of available solutes. A collinear structure in the bivariate plots could be used to infer conservative behavior. These data are used to construct a correlation matrix followed by principal components analysis (PCA) to extract eigenvectors and eigenvalues. The eigenvectors form the basis for a new Euclidean space, U-space. The new variables called the principal components (PC) are the coordinates in the U-space. The correlated chemical variables are projected orthogonally (by multiplying it with the eigenvectors) into U-space. Potential end-members, obtained from independent information, are then projected into this space and screened to see if they circumscribe the data. The number of eligible end-members is equal to the number of the PCs plus one, and they must: (1) form the polygon enclosing the streamflow concentrations data cloud in U-space; (2) be closest to this cloud in U-space.

Following the selection of the end-members, the chosen EMMA model is used to back-calculate the standardized stream water values. The retained PCA orthogonal projection coefficients and end-members are substituted into generalized equations for mixing model to derive flow path solutions (hydrograph separation). Last step is multiplying results of hydrograph separation (fractions) by original solute concentrations of end-members to reproduce streamflow chemistry for conservative solutes. A high correlation between series of modeled and measured concentrations of tracers in stream water indicates the adequacy of an obtained mixing model. Both successful and unsuccessful applications of EMMA may lead to testable hypotheses.





### 3.2. ECOMAG model

ECOMAG (ECOlogical Model for Applied Geophysics) is the process-based, semi-distributed RR model (Motovilov et
al., 1999). The spatial structure of the ECOMAG model splits watershed into sub-basins based on topography, river network
structure, soil and vegetation type, landuse, and variability of climate characteristics. Main ECOMAG model equations were
adopted from the full spatially distributed model (Kuchment et al., 1983) by neglecting secondary terms and spatial aggregation
at subbasin scale. Daily resolution timeseries of precipitation, air temperature and humidity are used as inputs. Computation
of river basin hydrological response described by two main phase: for each sub-basin calculate the effective precipitation and
then routing it through the river network. Runoff from sub-basin is calculated as sum of three components: Horton overland
(surface) flow (when rainfall has exceeded infiltration capacity and depression storage capacity), soil flow (sum of runoff from
horizon A and B) and groundwater outflow (baseflow and infiltration from horizon B). At warm period precipitation is partially
infiltrated and move along the hillslopes (over impermeable surface) as interflow. Excess water produce surface flow and
move downslope towards the drainage network. The rest of the water that has not been fed rivers as lateral or surface flow can
be evaporated or percolated into deep aquifers. Within cold and mid-season periods model describe snowpack evolution and
soil freezing - thawing cycle.

### 3.3. SWAT model

Soil and Water Assessment Tool (SWAT) is a semi-distributed hydro-ecological model (Arnold et al., 1993, Neitsch et
al., 2011). Model use the net of Hydrologic Response Units (HRU) as the base elements to model main hydrological cycle
processes - infiltration, evaporation, runoff generation, soil hydrothermic and snow cover dynamic. Daily volume of
precipitation excess and surface runoff calculate by empirical SCS CN method. The kinematic approximation is used for
channel routing. Daily rainfall, maximum and minimum air temperature, solar radiation, relative air humidity and wind speed
are the inputs to the model. SWAT use air temperature to define the input precipitation as rain or snow. Intercepted by canopy
part of precipitation is evaporated. Then simulation is performed as two successive land and routing phases. The snow thawing
is linearly relation with snow cover depth and fraction of watershed area and air temperature. At the top of soil profile
precipitated water can either flow overland or infiltrate into underlying stratums if they temperature is positive, field capacity
of upper layer is exceeded and underlying soil layer is not saturated. Water may partially evaporated from and turn into lateral
or groundwater runoff component inside soil profile. The dynamic of lateral flow runoff component is calculated using
saturated conductivity of each soil layer. Percolated through soil column water proportionally divided between unconfined and
the deep aquifers. Potential evaporation is used for calculate the water exchange between groundwater and bottom of soil
profile.





### 3.4. HBV model

The conceptual HBV model was developed by the Swedish Meteorological and Hydrological Institute (SMHI)
(Bergstrom, 1976). Model consists of three main modules: snowmelt and snow accumulation, soil moisture and effective
precipitation routine, runoff transformation to catchments outlet. The measured precipitation is supposed to be snow if the air
temperature is lower than threshold temperature TT; otherwise, precipitation appears as rain. Simulated snowpack volume can
be adjusted with correction factor $S_{CF}$. Degree-day method is used for snowmelt calculation. Groundwater recharge and actual
evaporation are simulated as functions of actual soil storage. Available for runoff generation amount of water calculates as
ratio of actual soil moisture SM to field capacity FC and the power coefficient Beta - $(SM\ FC^{-1})^{Beta}$. Model firstly recharges
the upper storage and then the lower storage using percolation parameter PERC. Surface flow appears when the upper storage
capacity exceeds the certain threshold. Potential evapotranspiration (PET) is input to the model, which can be tuned by Cet
parameter. Actual evaporation is equal to PET when $SM\ FC^{-1}$ is higher than LP parameter; otherwise, a linear reduction is
used. The runoff is calculated as the sum of three linear outflow: surface flow $Q_0$, interflow $Q_1$ and baseflow $Q_2$ with three
correspondent recession coefficients $K_0$, $K_1$ and $K_2$. A transformation function with the triangular weighting parameter
MAXBAS (Seibert and Vis, 2012) is used for smoothing the total runoff to obtain discharge at the outlet.

### 4. End-members mixing analysis results

EMMA analysis has been performed at Medvezhy catchment for the period 2015-2017, and at Elovy catchment for the
period 2012-2014. Bivariate solute plots have been constructed for all pairs of chemical indicators using the stream water
samples for whole period of observations. The diagrams exhibiting collinearity and linear trends were used to define tracers
with conservative behavior. Stronger linear trends between pair solutes are observed at Medvezhy catchment, and weaker
linear trends are observed at Elovy catchment, $R^2 < 0.5$ (Tab. 1).

For the Medvezhy catchment, PCA was performed for the matrix of chemical indicators including six solutes as TDS,
DOC, $HCO_3^-$, $SO_4^{2-}$, $Ca^{2+}$, and $Mg^{2+}$. First two PC explain more than 92 % of the total variability of these data (Tab. 2).
Analysis of PCA-model residuals against measured value for individual solute at two-dimensional mixing subspace is shown
in the Fig. 2a. Random pattern for each solute indicates that these six tracers could be considered as conservative and can be
used for the adopting the mixing model with three end-members.

For the Elovy catchment, the PCA models at the 2 and 3 dimensionality are fault, since the cumulative dispersion of
the first three PC does not exceed 88 %. Four end-members (EMs) mixing model is required to successful represent the runoff
generation in the Elovy Creek (Gubareva et al., 2015, 2018, 2019). The $HCO_3^-$, $Mg^{2+}$, $NO_3^-$, $Na^+$ tracers provide the 95 %
cumulative proportion explained by first three principal component. The hypothesis about tracer conservatism is confirmed by
randomness structures of residuals demonstrated in Fig. 2b.



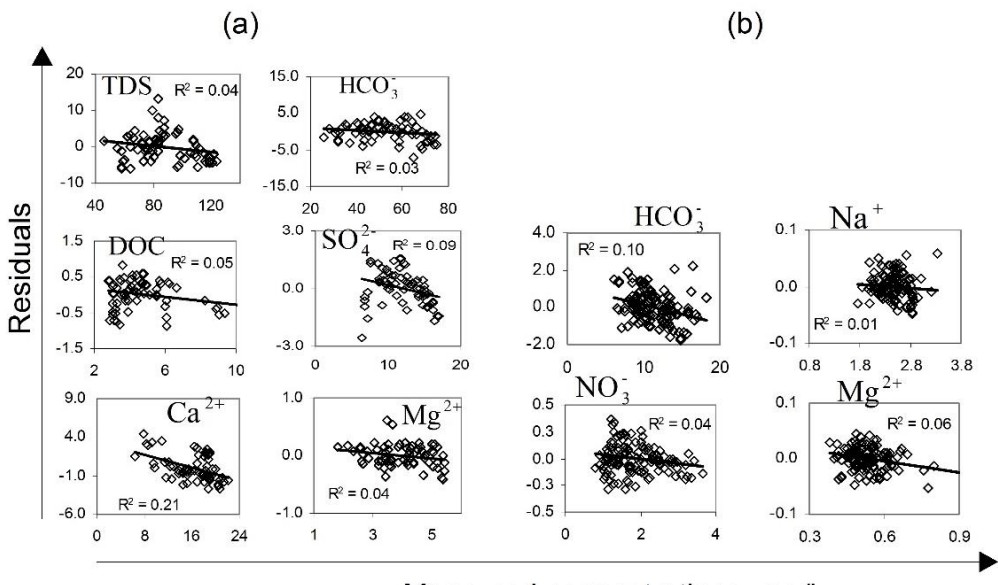

**Figure 2: The residuals plots for 2D mixing subspace: a) Medvezhy catchment; b) Elovy catchment.**

**Table 1** Pairwise correlations of the concentrations of solutes, ranked by R2 (n is the number of samples).

| $R^2$ | Medvezhy catchment, n=85 | Elovy catchment, n=133 |
|---|---|---|
| > 0.71 | $TDS - HCO_3^-$, $SO_4^{2-} - Mg^{2+}$, $HCO_3^- - Mg^{2+}$, $TDS - Mg^{2+}$ | – |
| 0.7-0.61 | $DOC - SO_4^{2-}$, $HCO_3^- - SO_4^{2-}$, $SO_4^{2-} - Ca^{2+}$, $HCO_3^- - Ca^{2+}$, $TDS - Ca^{2+}$, $Ca^{2+} - Mg^{2+}$ | – |
| 0.6-0.51 | $TDS - SO_4^{2-}$, $DOC - Mg^{2+}$ | – |
| 0.5-0.41 | – | $HCO_3^- - Mg^{2+}$, $HCO_3^- — TDS$, $SO_4^{2-} - NO_3^-$ |
| 0.4-0.3 | – | $NO_3^- - HCO_3^-$, $NO_3^- - Na^+$, $HCO_3^- - Na^+$, $Na^+ - TDS$ |

**Table 2** The percent of variance explained by PC (U) for individual catchments.

| Catchment | Tracers | U1 | U2 | U3 | U4 | U5 | U6 |
|---|---|---|---|---|---|---|---|
| Medvezhy | TDS, DOC, $HCO_3^-$, $SO_4^{2-}$, $Ca^2$, $Mg^{2+}$ | 83.6 | 8.6 (92.2) | 4.4 (96.7) | 2.1 (98.7) | 0.9 (99.6) | 0.4 (100) |
| Elovy | $HCO_3^-$, $Mg^{2+}$, $NO_3^-$, $Na^+$, TDS, $SO_4^{2-}$ | 54.8 | 24.1 (78.9) | 8.9 (87.8) | 5.2 (93.0) | 4.6 (97.6) | 2.4 (100) |

*Values in parentheses indicate the cumulative percent variance in stream water chemistry explained by multiple retained principal components.



Mixing diagrams have been constructed for each stream by orthogonally projecting stream water matrix of
conservative tracers into U-space. Tracers averaged concentrations of EMs were also projected into the same U-space using
eigenvectors. The 2D mixing diagram at Medvezhy catchment is presented in Fig. 3a. River samples are located in the mixing
subspace of tracers <U1, U2> and represented by three sources. The mixing diagram at Elovy catchment represents the cloud
of river samples, which are located in the three-dimensional mixing subspace <U1, U2, U3> and defined by four sources
(Fig. 3b). In both cases, the EMs-based shapes (triangle or tetrahedron) enclose most of the river samples in the U-space. This
geometrically confirms the possibility of representing the composition of each river sample as a mixture of water sources, and
generally verifies the mixing model.

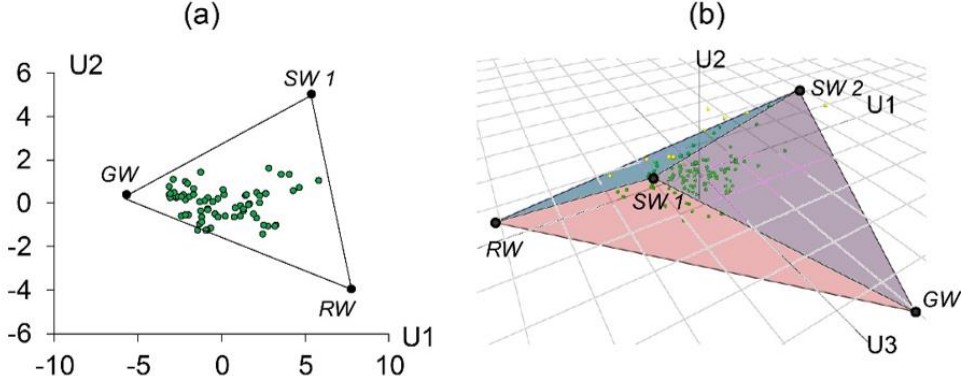

**Figure 3: Mixing diagrams in the U-space: a) Medvezhy catchment; b) Elovy catchment. RW – rainwater, GW – ground water,**
**SW1, SW2 – soil water.**

        The identification and interpretation of EMs for the Medvezhy catchment is unambiguous. Two EMs (GW and SW1)
are related with ground and soil water, and the third one with rainwater (RW). The GW is related with streamflow samples
taken during the low flow period and represents the baseflow component of the runoff. Samples of SW1 were taken from the
upper organic horizon of soil and can be interpreted as lateral (or inter soil) flow. RW reflects quick overland runoff that reach
the drainage network almost without chemical transformation. The mixing subspace at Elovy catchment evolved by four EMs.
First three of them (RW, GW and SW1) are the same as for Medvezhy, the fourth, SW2 is associated with soil mineral
component sampled from lysimetric water at the bottom of the steep hillslope from fir-spruce forest soil with lower rate of
organic matter destruction.

        The EMMA runoff components and measured discharges were used to build the relationship: GW = f (Q) and SW1
= f (Q) at Medvezhy catchment; GW = f (Q), SW1 = f (Q) and SW1 + SW2 = f (Q) at Elovy one (Fig. 4). The fractions of RW
(for both stream) and SW2 (for Elovy), which are weakly related with measured discharges, are determined as the remainder
from the water balance equation. These relating curves were used to calculate the daily streamflow composition series using
the measured hydrograph.





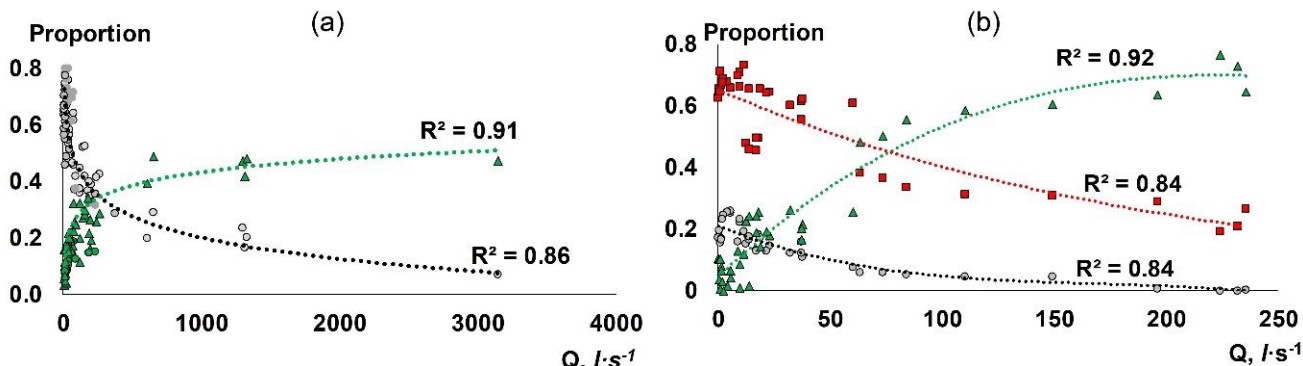

**Figure 4: Streamflow components rating curves: a) Medvezhy catchment; b) Elovy catchment. GW - black, SW1 - green, and SW1 + SW2 – red.**

## 5. Hydrological simulation results

All used models were calibrated independently. Hydrological simulations were made continuously (winter season included) with daily time step and one year warm-up period. The details on calibration and comments on parameters values of 
the models placed in the appendix. Observed and simulated hydrographs for typhoon-induced flood events of the 2012 and 2016 are given in (Fig. 5). Model efficiency assessed using common goodness-of-fit measures (determination factor $R^2$, Nash and Sutcliffe (NSE) efficiency (Nash and Sutcliffe, 1970), and relative bias (BIAS, %) against the measured discharge at catchments outlet (Tab. 3). NSE is categorized as "very good" when its value > 0.75 and "unsatisfactory" when its value < 0.5, interim ranges (0.75 > NSE > 0.65 and 0.65 > NSE > 0.5) are defined as "good" and "satisfactory" correspondingly. In 
reference to BIAS values, simulation results are assumed as unacceptable if |BIAS| > 25 %, "satisfactory" at 15 % <|BIAS| < 25 %, "good" for 15 % <|BIAS| < 10 % and "very good" at |BIAS| < 10 % (Moriasi et al., 2007).

According to these criteria, more complex and sophisticated models give the better results for the both catchments and ECOMAG outperforms SWAT in most cases. HBV gives good results for the period of high flow but demonstrates poor performance for low flow (0.01–1.0 mm day[-1]) periods.

## 250 6. Hydrograph separation results and discussion

Dynamics of runoff composition for the years with significant flood events is presented in (Fig. 6). All RR models demonstrated that two neighboring small catchments significantly differ in mutual dynamics of the runoff fractions. Geochemical analysis confirmed the differences in runoff generation processes at the both studied catchments. For the Medvezhy catchment, the mixing model gave a typical 3-sources stable structure and reliable estimates of the fractions. The 
fraction of soil runoff for this catchment is relatively small. Opposite, the Elovy catchment soil fraction is more significant





against surface and ground ones. Here the more complex geology, higher elevation range within smaller catchment area, and at least two apparent soil-vegetation belts lead to apply a more complex 4-sources EMMA model (two soil fractions).

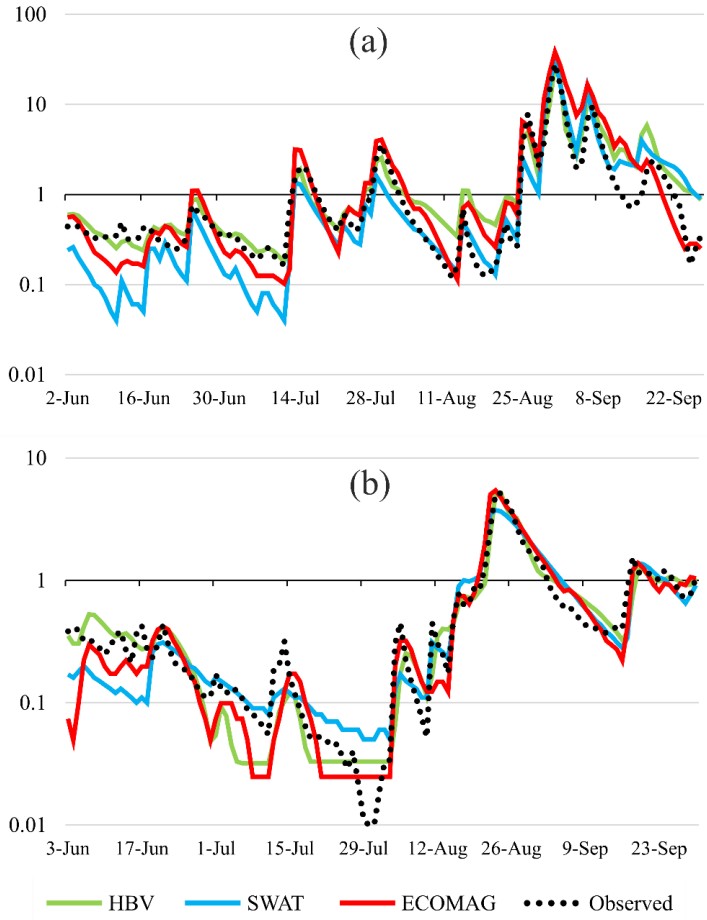

**Figure 5: Examples of measured and simulated hydrographs (mm/d): a) Medvezhy catchment, 2016 b) Elovy catchment, 2012.**

We would like to highlight the two main aspects concerning the correctness of further comparison of runoff composition provided by different models. First, it is difficult to reconcile the interpretations of modeled runoff fractions. The hydrological models derive runoff components from a hydro physical and hydraulic characteristics of each flow paths. Their results are conditioned by the different basic concepts used, always more or less speculative. On the other hand, the tracer separation model is based on the interpretation of measured hydrochemical data related to water residence time, geochemical features of rocks and interaction intensity with matrix. The runoff fractions in different models, denoted by the same terms, may not have the same sense. The problem of compatibility and reconciling of different interpretations of the river flow structure is one of the most complex problems of the runoff theory and is beyond of the scope of this study.




**Table 3** Goodness-of-fit characteristics for runoff simulations with different models.

| Model | Medvezhy catchment | | | | Elovy catchment | | | |
|---|---|---|---|---|---|---|---|---|
| | Year | $R^2$ | NSE | BIAS, % | Year | $R^2$ | NSE | BIAS, % |
| ECOMAG | 2015 | 0.92 | 0.91 | -5 | 2012 | 0.93 | 0.91 | -0.3 |
| | 2016 | 0.92 | 0.9 | 8 | 2013 | 0.87 | 0.8 | 13 |
| | 2017 | 0.91 | 0.87 | -6 | 2014 | 0.84 | 0.83 | 4 |
| | 2015-2017 | 0.92 | 0.9 | 4 | 2012-2014 | 0.9 | 0.89 | 10 |
| SWAT | 2015 | 0.94 | 0.88 | -20 | 2012 | 0.9 | 0.9 | -4 |
| | 2016 | 0.89 | 0.85 | 6 | 2013 | 0.82 | 0.81 | -1 |
| | 2017 | 0.69 | 0.67 | -12 | 2014 | 0.93 | 0.88 | -13 |
| | 2015-2017 | 0.9 | 0.86 | -1 | 2012-2014 | 0.86 | 0.86 | -6 |
| HBV | 2015 | 0.35 | 0.35 | -8 | 2012 | 0.96 | 0.96 | 1 |
| | 2016 | 0.92 | 0.91 | 18 | 2013 | 0.83 | 0.83 | -6 |
| | 2017 | 0.56 | 0.53 | -3 | 2014 | 0.64 | 0.62 | 4 |
| | 2015-2017 | 0.91 | 0.91 | 12 | 2012-2014 | 0.88 | 0.88 | -0.3 |

Second, the comparison may be complicated by relatively low accuracy and instability in daily runoff fractions, "measured" by the EMMA. This is due to the both well-known limitations of the EMMA (incomplete conservatism of tracers and variability of sources), and the limited number of samples leads to significant dispersion in statistical relationships used to

calculate daily series of the runoff fractions. Concerning this problem, the data aggregation in longer time intervals should allow to mutually compensate the random errors and to smooth the instability in daily runoff fractions. Therefore, the range of data aggregation intervals (season, month and pentad) are applied to find a reasonable data generalization period ensuring results clarity.

The further benchmark analysis is based on the ranking of runoff components proximity simulated by rainfall-runoff

models to the hydrograph decomposition by EMMA, regarded as identification model. Each RR models used (ECOMAG, HBV, SWAT) provides three runoff fractions related to surface, soil and ground water. The EMMA mixing model separated the Medvezhy catchment runoff to the three fractions correspond to the runoff structure of hydrological models; to match three runoff components structure in other models two EMMA soil fractions are summarized for the Elovy catchment.

At the first step, the most generalized seasonal (June-September) data aggregation (Tab. 4) was examined. The total

seasonal runoff volume for the Medvezhy catchment is reproduced more accurate: by ECOMAG in 2015 and by HBV in 2017; in 2016 SWAT and ECOMAG demonstrate almost identical proximity. The seasonal runoff composition, obtained from EMMA, was captured best using ECOMAG for the dry seasons as 2015 and 2017. However, in the high water season of 2016, it was impossible to choose the best hydrograph separation (as the closest to reference EMMA) among the three simulated variants discussed.




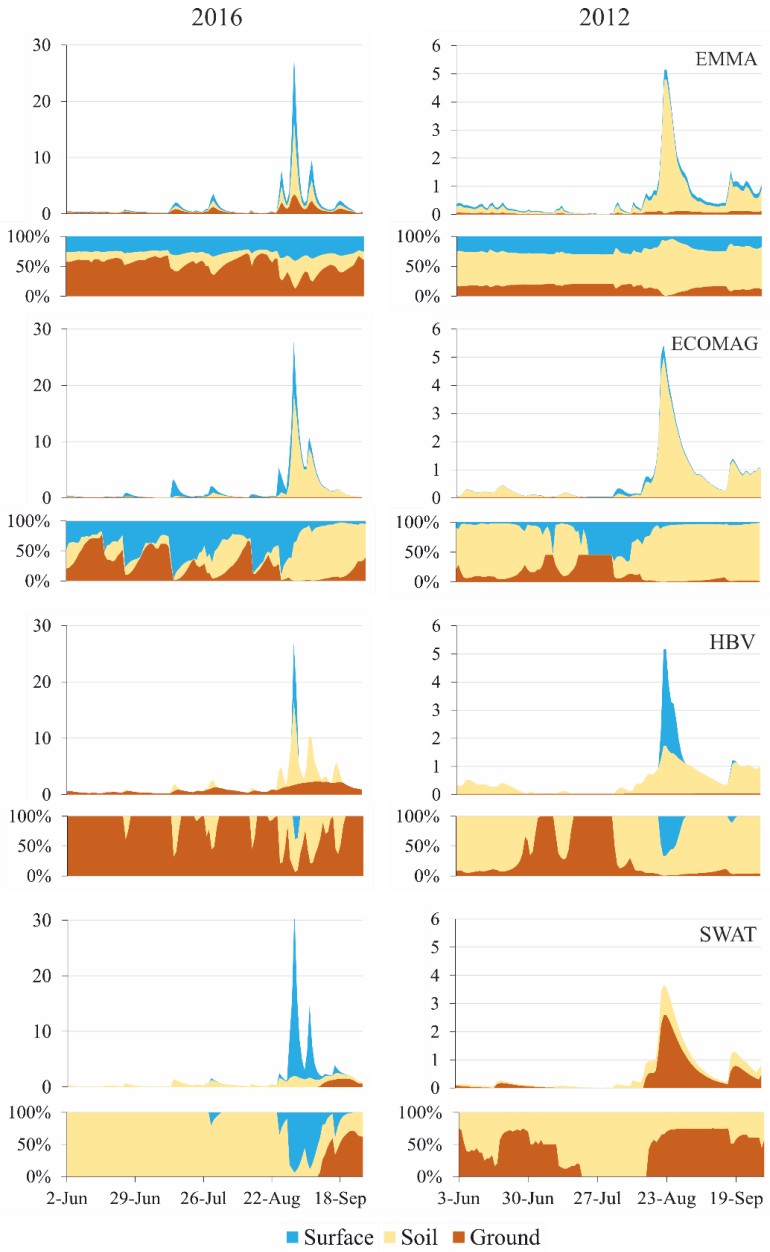

**Figure 6: Examples of calculated absolute (mm day⁻¹) and relative fraction of daily runoff components for Medvezhy catchment (left panel - 2016) and Elovy catchment (right panel - 2012)**

In most cases of the Elovy catchment runoff separation (Tab. 4), HBV seems slightly better than ECOMAG, and it was

most successful in the estimating of groundwater component. SWAT for the both catchments provided the runoff composition essentially different against EMMA and greatly underestimated the Elovy catchment total seasonal runoff volume. Thus, the seasonal data aggregation is not optimal for unambiguous selection of the best simulation model, at least given the available data. Next, the monthly data aggregations were used to rank rainfall-runoff models according to better approximation to





reference EMMA. compare monthly data aggregation the results of the three runoff simulations were compared with 12

observed monthly runoff volumes (four months in three years) and 36 its fractions estimate obtained by EMMA (Fig. 7). For each case the simulation giving the closest approximation was fixed, then the total number of the best results was calculated.

**Table 4** Seasonal runoff components obtained from rainfall-runoff models and results of hydrograph separation by EMMA.

| Source | Medvezhy catchment | | | | Elovy catchment* | | | |
|---|---|---|---|---|---|---|---|---|
| | SWAT | HBV | ECOMAG | EMMA | SWAT | HBV | ECOMAG | EMMA |
| | 2015 | | | | 2012 | | | |
| Surface, % | 0 | 0 | 43.5 | 25.6 | 0 | 22.1 | 7.7 | 15.1 |
| Soil, % | 97.8 | 2.8 | 29.9 | 14.6 | 35.3 | 72.3 | 89.3 | 75.5 |
| Ground, % | 2.2 | 97.2 | 26.6 | 59.8 | 64.7 | 5.6 | 3 | 9.4 |
| Total, mm | 26.9 | 43.9 | 34.6 | 37.2 | 64.6 | 74.1 | 78 | 74.1 |
| | 2016 | | | | 2013 | | | |
| Surface, % | 57 | 7.8 | 32.1 | 34.4 | 0 | 5 | 3.5 | 24.4 |
| Soil, % | 33.3 | 43.7 | 63 | 33 | 36.2 | 85.9 | 94.2 | 66.3 |
| Ground, % | 9.7 | 48.5 | 4.9 | 32.6 | 63.8 | 9.1 | 2.3 | 9.3 |
| Total, mm | 211.3 | 233.4 | 213.3 | 195.2 | 62.2 | 76.6 | 100.3 | 85.6 |
| | 2017 | | | | 2014 | | | |
| Surface, % | 0.7 | 0 | 58.3 | 27.3 | 0.1 | 1 | 3.8 | 23.8 |
| Soil, % | 99.3 | 13.3 | 22.9 | 18.3 | 45.4 | 86.2 | 92.8 | 63.2 |
| Ground, % | 0 | 86.7 | 18.8 | 54.4 | 54.5 | 12.8 | 3.3 | 12.9 |
| Total, mm | 35 | 38.1 | 37 | 38.7 | 49.4 | 73.7 | 71 | 66.6 |
| | 2015-2017 | | | | 2012-2014 | | | |
| Surface, % | 44.1 | 5.7 | 35.8 | 32.1 | 0 | 9.3 | 4.9 | 21.2 |
| Soil, % | 47.8 | 34.4 | 54.8 | 28.4 | 38.4 | 81.5 | 92.3 | 68.4 |
| Ground, % | 8.1 | 59.9 | 9.4 | 39.5 | 61.6 | 9.2 | 2.8 | 10.4 |
| Total, mm | 273.1 | 315.5 | 284.9 | 271.2 | 176.2 | 224.3 | 249.3 | 226.3 |

*Soil fraction for Elovy catchment obtained by EMMA is the sum of the SW1 (soil organic) and SW2 (soil mineral) parts,
see section 4 for details

For the Medvezhy catchment, the closest proximity of the ECOMAG and EMMA runoff composition is more obvious (Fig. 7). Comparison of monthly runoff volumes showed the approximate parity between HBV and ECOMAG - 7 versus 5 cases, respectively (SWAT lose the all cases). ECOMAG runoff composition is close to EMMA results in 22 out of 36 cases; it is followed by the HBV with 11 cases, while SWAT is successful only in 3 cases.

Opposite, for the Elovy catchment, the runoff compositions of the HBV and ECOMAG models is similar close to the EMMA hydrograph separation (Fig. 7). For runoff volumes of the Elovy creek both ECOMAG and HBV score 5 cases, and SWAT remaining two. The Elovy monthly runoff fractions provided by HBV and ECOMAG are close to EMMA estimates in 16 cases for every, and SWAT - in 4 cases. An advantage of the HBV model clearly appears in estimation of the groundwater





fraction in streamflow, as well as an advantage of the ECOMAG model - in estimation of the surface one. The SWAT model
hydrograph decomposition based on monthly analysis looks uncompetitive.

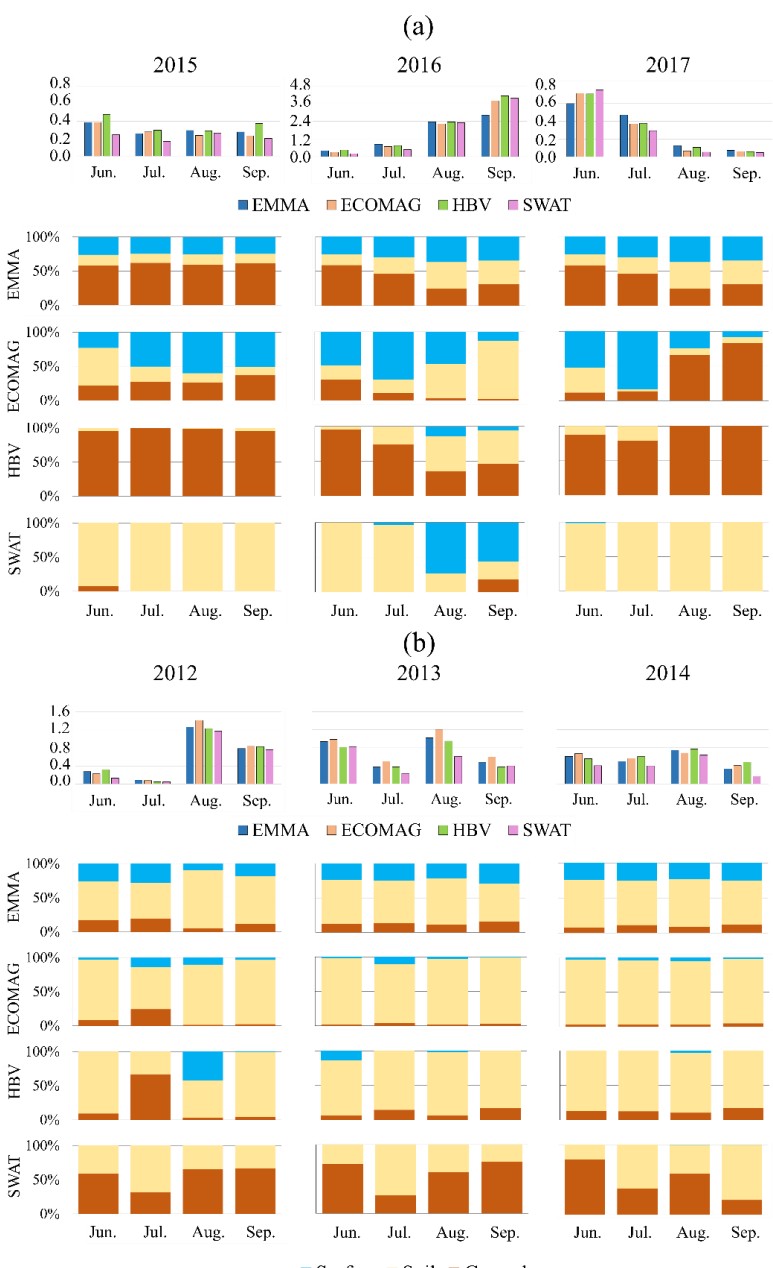

**Figure 7: Comparison of calculated monthly absolute (mm day⁻¹) and relative (%) proportions of runoff components: a – Medvezhy catchment, b – Elovy catchment.**




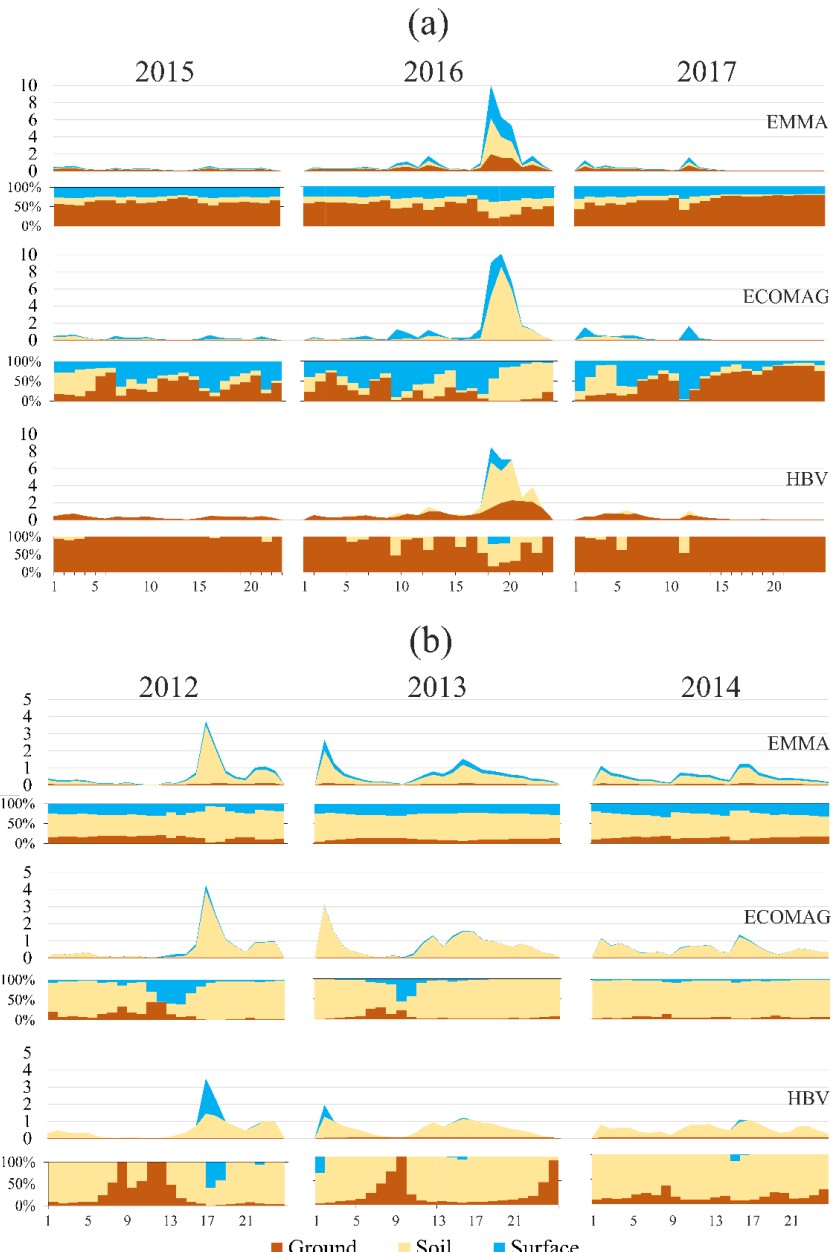

**Figure 8: Comparison of calculated pentade absolute (mm day⁻¹) and relative (%) proportions of runoff components: a – Medvezhy catchment, b – Elovy catchment.**

Based on results of five days data aggregation comparison (Fig. 8), the advantage of the ECOMAG model for the
Medvezhy catchment becomes more pronounced. The ECOMAG runoff fractions dynamic is much similar to the EMMA decomposition. However, for the outstanding 2016 flood event HBV results are closer to the EMMA, while the ECOMAG has given significantly different runoff composition. For the Elovy catchment it is hard to choose with confidence the best





approximation to EMMA decomposition from results of ECOMAG and HBV models. It should be noted along, that three years (2012-2014) forming the observation series on Elovy catchment are similar in terms of volume and regime of runoff, in contrast to 2015-2017, included in the observation series on Medvezhy catchment. The extreme event similar to the flood of 2016 has not been measured on the Elovy catchment.

In accordance with the comparing results, the ECOMAG model yielded the best agreement with EMMA, HBV comes in second, and SWAT model (despite of it outperform a bit the HBV by NSE efficiency) sink to the bottom rank position. Competitive advantage of the ECOMAG model is provided by proportions comparability of all three fractions in the runoff composition and the mutual dynamics of their values as well. However, while the surface runoff fractions in ECOMAG are in good agreement with EMMA, the subsurface water components estimations differ markedly. In some cases, the HBV model is able to reproduce the EMMA ground runoff fraction pretty accurate, but rather the overall pattern over relatively long (seasonal) intervals, not its detailed behavior. SWAT at all the times showed more difference with EMMA than other models.

All the above noted inferences about of the representation features of runoff generation in various models have been formulated only preliminarily based on limited data. It should be concluded that monthly and synoptic (five/ten days) data aggregation are the scales at which the seasonal runoff dynamics is well suited to clearly compare the rainfall-runoff models with representation of flood events in aggregated view.

## 7. Conclusions

The article demonstrates the results of comparing the simulated runoff hydrographs and its fractions, obtained by calibration of three different rainfall-runoff models (SWAT, HBV, ECOMAG), with observed runoff hydrographs and the dynamics of its fractions obtained by EMMA approach. For the study we used in situ observations in two typical small experimental catchments of the mountain-taiga region in the south of Pacific Russia. Both the rainfall-runoff and tracer models provide unequivocal evidence that two neighboring small catchments significantly differ in structure of runoff generation processes related to difference in the geological and landscape structure. It is shown that compliance with EMMA results may give auxiliary useful information for validation of hydrological modeling results besides of use the standard efficiency criteria reflected proximity of simulated and modeling values of runoff.

Summarizing the above, we can give a preliminary answer to the question motivated this study. Comparison of runoff composition obtained with direct and inverse modeling can rank the rainfall-runoff models for completeness of the runoff generation processes physical description. At the same time, the problem of reaching a detailed correspondence of the runoff composition dynamics is remain an urgent challenge. This requires a development for reference classifications of water masses types within a river basin and knowledge of runoff generation mechanisms taking into account regional specifics. Also, it is necessary both the observation techniques improvement and modeling algorithms, and the accumulation of an extensive dataset on hydrograph separation. All obtained conclusions are preliminary based on limited data of two analyzed objects. The main





perspectives of the research require progress towards development in methodology, terminological and conceptual
compatibility of interpretations of streamflow composition behind various approaches.

## Appendix A. ECOMAG hydrological simulation results

The model was used by the authors for large watersheds of lowland rivers in European Russia (Motovilov, 2016; Kalugin,
2019), arctic rivers (Gelfan et al, 2020), as well as the Amur River (Kalugin and Motovilov, 2018) and for medium-sized semi-
mountain river (Kalugin, 2021). In this study, the model was applied to the scale of an experimental catchment in Russia for
the first time. The average area of model units for the studied catchments varies within 0.3–0.5 km$^2$. In the model, the soil and
groundwater flow was described by the Darcy equation, and the surface flow was described by the kinematic wave equation.
In conditions of high soil moisture, the actual evaporation was equal to the potential, and then it decreased linearly to zero as
the soil moisture decreased to wilting point. Potential evaporation was estimated according to the Dalton method. The
snowmelt rate is calculated using the degree-day method. Initial parameters values were assigned from available measurements
and databases. During calibration the ratio between the initial and optimized parameter values is fixed (Gelfan et al., 2015).
The values of the main calibrated parameters are presented in (Tab. A5).

**Table A5** Values of ECOMAG calibrated parameters values.

| Parameter | Short name | Medvezhy | Elovy |
|---|---|---|---|
| Coef. of vertical saturated hydraulic conductivity | GFB | 8.3 | 6.5 |
| Coef. of horizontal saturated hydraulic conductivity | GFA | 1 | 10 |
| Soil evaporation coefficient | EK | 0.75 | 0.8 |
| Baseflow constant, mm day$^{-1}$ | GROUND | 0.009 | 0.0001 |
| Coef. of snowmelt intensity, mm day$^{-1}$ °C | ALF | 0.28 | 0.45 |
| Critical air temperature snow/rain, °C | TCR | 0.5 | 0.5 |
| Snowmelt air temperature, °C | TSN | 0.1 | 0.1 |
| Air temperature gradient, °C 100 m$^{-1}$ | TGR | −0.6 | −0.6 |
| Precipitation gradient, mm 100 m$^{-1}$ | PGR | 0 | 0 |

In the previous experience of applying ECOMAG for the whole Upper-Ussuri River basin (Motovilov et al., 2018) it was
found that the most sensitive parameters were EK and GFB. To simulate the hydrological response to intensive rainfall of the
small studied mountain catchments with steep hillslopes and high horizontal soil saturated hydraulic conductivity two
additional parameters GFA and GROUND were added into the calibration procedure. Calibrated values for vertical and
horizontal saturated hydraulic conductivity (GFB and GFA) demonstrated unexpected big difference for two near located
catchments. The GFA value for the Elovy Creek was an order of magnitude greater in comparison with the Medvezhy Creek,
and the GFB parameter, on the contrary, was lower. Soil evaporation coefficient (EK) is slightly higher for Elovy Creek. The
baseflow constant parameter (GROUND) was an order of magnitude smaller than the values obtained in the previous study





for its major watershed (Motovilov et al., 2018) and differed by almost two orders of magnitude for the two studied catchments. All these facts indicate the difference in the runoff generation and distinctive geological and geomorphological conditions of

the studied objects. Given the study of the runoff over the summer period, the model was insensitive to the parameters of the intensity of snowmelt (ALF) as well as the temperature threshold for the precipitation phase (TCR) and snowmelt (TSN). For both studied catchments, the air temperature gradient (TGR) was normal value, and the precipitation gradient (PGR) was not used because the precipitation in the model was determined from the data of two automatic weather stations, which are representative of the catchments.

**Appendix B. SWAT hydrological simulation results**

ArcSWAT 2012 GIS interface was used for preparing simulation and model calibration. Model HRUs are subbasins of area 1–3 km$^2$. Potential evaporation computed by Penman–Monteith method. Channel routing by variable travel time method. Set of calibrated parameters and its values are presented in (Tab. B6). Values of CN2 (runoff curve number) of both basins correspond to group "A" of high infiltration capacity soils (USDA, 1956). Calibrated values of evaporation compensation

factor (ESCO) from soil profile allow trees roots extract water from all soil layers for evapotranspiration. Parameters for groundwater simulation (DEP_IMP, ALPHA_BF, GW_DELAY and GWQMIN) were specified during model calibration.

**Table B6** SWAT calibrated parameters values.

| Parameter | Short name | Medvezhy | Elovy |
|---|---|---|---|
| SCS runoff curve number for moisture condition II | CN2 | 35.0 | 35.0 |
| Roughness coefficient for overland flow | OV_N | 30.0 | 0.01 |
| Evaporation compensation from soil | ESCO | 0.1 | 0.46 |
| Travel time of lateral flow, days | LAT_TTIME | 3.5 | 7.7 |
| Depth of the impervious layer, m | DEP_IMP | 4.25 | 5.1 |
| Baseflow recession constant | ALPHA_BF | 0.25 | 0.13 |
| Time to reach the groundwater, days | GW_DELAY | 1.5 | 1.55 |
| Recharge of deep aquifer coefficient | RCHRG_DP | 0.55 | 0.24 |
| Threshold for return flow to occur, mm | GWQMN | 50 | 0.0 |
| Capillary rise coefficient | GW_REVAP | 0.2 | 0.2 |
| Threshold for GW_REVAP to occur, mm | REVAPMN | 25 | 0 |
| Slope length for lateral subsurface flow, m | SLSOIL | 48 | 59 |

High values of OV_N for the Medvezhy creek means during the flood events most of the rain water quick infiltrate into the soil and reach the catchment drainage network as interflow through the system of subsoil drains (Gerrard, 1981; Bugaets et al., 2019; Gartsman et al., 2020). The roughness should be considered as total hillslope flow resistance (Bugaets et al., 2018, 2021). In case of Elovy creek the model is insensitive to OV_N parameter which means that fraction of surface flow is negligible. The principal difference is between ESCO and RCHRG_DP parameters. It means that Elovy basin evaporate more





water and Medvezhy loss significant amount of water from soil profile for recharge deep aquifer (losses). The model systematically underestimates the maximum runoff for the both catchments. This fact can be explained by two facts: first is underestimation of CN value for the soil saturation condition, and the second is overestimation of rainfall losses for the vegetation to interception during periods of heavy rains (> 100 mm day$^{-1}$).

**Appendix B. HBV hydrological simulation results**

410       HBV-light standard version was used, calibration was performed manually using user-interface developed by (Seibert and Vis, 2012). PET was calculated by Penman-Monteith method on daily basis. Obtained model parameters (Tab. C7) clearly demonstrate the difference in FC, Beta, PERC, recession and Cet factors for investigated catchments. The instream routed MAXBAS is in the same range. The obtained soil profile related parameters lead to deeper and more permiable soil for Medvezhy. Difference in Cet parameter is low and can be explained by watershed hillslope aspect. Higher values of recession

coefficients for Medvezhy correspond to fast lateral flow. Special attention draws the recession coefficient K2 for Elovy creek. The value of K2 actually mean that this part is not presented for this small catchment. Used model version does not allow deep aquifer losses, so in case of long-term calculation, the near zero value of K2 will lead to accumulate water in the low storage and significant bias in modeling result.

420                                   **Table C7** Values of HBV calibrated parameters.

| Parameter | Short name | Medvezhy | Elovy |
|---|---|---|---|
| Max. soil storage content, mm | FC | 350 | 150 |
| PET limit | LP | 0.2 | 0.58 |
| Recharge parameter | Beta | 1.7 | 2.7 |
| Max. percolation rate, mm day$^{-1}$ | PERC | 2.6 | 1.6 |
| Upper zone limit, mm | HL | 28 | 45 |
| Recession coef. | K0 | 0.99 | 0.16 |
| Recession coef. | K1 | 0.40 | 0.03 |
| Recession coef. | K2 | 0.13 | 0.0001 |
| Length of triangular weighting function, days | MAXBAS | 1.8 | 2.9 |
| PET correction factor, 1 °C$^{-1}$ | Cet | 0 | 0.03 |

**Data availability**

Data sets for this study are under preparation for public release by the authors and can be freely obtained from andreybugaets@yandex.ru



**Author contributions**

AB, TG, SL and AK are responsible for hydrological and hydrochemical modeling. SL, TG and LG performed data preparation. AB, BG, SL and VS provided data analysis. AB, SL and BG worked with development of the article structure. All authors took participation in final edit of the article.

**Competing interests**

The authors declare that they have no conflict of interest.

**Financial support**

*This research was funded by the Ministry of Science and Higher Education of the Russian Federation, themes 0149-2019-0003, AAAA-A19-119030790003-1, AAAA-A19-119102290002-3; by the Russian Foundation for Basic Research, research projects 19-05-00353 and 19-05-00326.*

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
