# Peer review of "Comparing the runoff decompositions of small testbed catchments: end-member mixing analysis against hydrological modelling"

_Hydrology and Earth System Sciences, 2021_

## Author Comment (AC1)

**I know I should start the review by summarizing the main characteristics of the paper, but I was unable to discern any. I broke off the review at page 5 because this paper is too carelessly prepared. The English is very difficult to understand, the HESS guidelines have been poorly adhered to. Furthermore, the paper has fundamental weaknesses.**

**The issues raised in the Introduction cannot be addressed by a study that relies on two small catchments located very close to each other.**

**The paper criticizes the use of empirical relations in hydrological modelling, then relies on empirical relationships itself.**

**Parts of the methodology are poorly and/or incompletely explained.**

**Given these problems, I do not think it is worthwhile to spend more time on this paper. I am sorry for this, but I am simply losing too much time struggling through the text.**

In fact, there is no benefit in responding to a review that refers to the first 5 pages of a large article. Nevertheless, the authors are grateful to the referee for his time spent. The key and useful remark of the reviewer should be considered as a criticism of the poor English language of the article. The authors accept all language and editorial comments and intend to make serious efforts to improve the text.

Obviously, these textual difficulties did not allow the reviewer to comprehend the main idea and the results of the study, although the materials presented in article, according to the authors, quite adequately reflect them. The key problem of validation and effective verification of hydrological models is well known and widely discussed in the professional community. The authors agree with the reviewer's statement "The issues raised in the Introduction cannot be addressed by a study that relies on two small catchments located very close to each other" and did not set themselves such a task. The purpose of this article was to present efforts, and some progress, to develop an approach to solving the problem of verifying hydrological models based on a comparison of the runoff sources composition estimated by solving direct and inverse modeling problems. This goal is clearly stated in the text and seems quite natural. From the reviewer's point of view, however, "Forward modeling and inverse modeling are completely different activities with very different goals." But any of the approaches to runoff modeling is aimed at the study of the same subject and the search for the same "truth". Or does the reviewer think that the results of any methodological approach should be considered separately, and it does not matter if each of them gives its own "different truth"?

The approach developed by the authors, of course, is not new, but it cannot be considered well developed and widely used. The author's research has elements of originality and is based on a unique (for the region) data set collected over many years of own field work. In particular, it shows that the very close location of two small watersheds does not prevent them from differing greatly in terms of runoff formation mechanisms and runoff component composition, i.e. to reflect so wide range of natural conditions. The results of the study have both purely scientific and applied value.

In conclusion, the authors once again express their gratitude for the recommendations regarding the English language of the article.

**Below are the comments that I was able to make.**

**English editing is needed. I found 4 grammatical errors in the first paragraph alone and stopped checking them after that because I do not have the time:**

**l.23: takes -> take**
**l.24: extrapolating -> extrapolated**
**l.26-27: challenge task -> a challenging task**
**l.50: ground flow -> groundwater flow**

Accepted, the text to be corrected.

**l.29: ...for validation flow pathways and residence times...: I do not understand what you are trying to say.**
**l.37-38: ...evolving model... top-down strategy: I do not understand.**
**l.42-43: ..the scale-dependency of HRU-based model ... small-scale physical laws...**
**The English is so warped I cannot understand this.**

Few references to facilitate reading terms:

1) P. Rodgers, C. Soulsby, S. Waldron, and D. Tetzlaff Using stable isotope tracers to assess hydrological flow paths, residence times and landscape influences in a nested mesoscale catchment https://hess.copernicus.org/articles/9/139/2005/hess-9-139-2005.html
2) Markus Hrachowitz and Martyn P. Clark. HESS Opinions: The complementary merits of competing modelling philosophies in hydrology https://hess.copernicus.org/articles/21/3953/2017/hess-21-3953-2017.pdf
3) Twenty-three unsolved problems in hydrology (UPH) – a community perspective. https://www.tandfonline.com/doi/full/10.1080/02626667.2019.1620507
4) V. K. GuptaI. Rodríguez-IturbeE. F. Wood (1986) Scale Problems in Hydrology: Runoff Generation and Basin Response https://link.springer.com/book/10.1007/978-94-009-4678-1
5) Wood, E. F.: Scaling behaviour of hydrological fluxes and variables: Empirical studies using a hydrological model and remote sensing data, Hydrol. Process., 9, 331–346, https://doi.org/10.1002/hyp.3360090308, 1995.
6) Wood, E. F., Sivapalan, M., Beven, K., and Band, L.: Effects of spatial variability and scale with implications to hydrologic modeling, J. Hydrol., 102, 29–47, https://doi.org/10.1016/0022-1694(88)90090-X , 1988.

**l.40: You claim that catchment hydrology is still very much empirical by quoting a single reference that is over 36 years old! This statement has no credibility at all.**

The references are checked and will be updated

**l.54-55: There are a number of examples...lack of suitable data sets. It is unclear to me what this means, but the second part seems to contradict the first part.**

The text will be edited to clarify that it is too early to talk about the wide spread of such studies due to lack geochemical data.

**l.58-59: How can you rank models based on processes?**

As indicated in the text of the article, we consider the results of hydrograph decomposition by the EMMA method as factual data on runoff sources. The various runoff simulation models are ranked according to how closely they reproduce the dynamics of the runoff sources while simulating the dynamics of the water discharge.

**l.60: ...based on solutions of direct or inverse task of modeling... Forward modeling and inverse modeling are completely differen activities with very different goals. Why are you using them as if they are similar?**

Essentially, we look for hydrological model that can reproduce a certain runoff composition (derived by EMMA). If it does, it will indicate that this model had better represent particular hydrological system.

**l.61-65: We can read the section headings, so there is no need to provide a table of contents. Instead, formulate the objective of the paper.**

The text will be checked along the edition process. This study is focus on comparison the catchment streamflow composition simulated with three well-known RR models against the hydrograph decomposition obtained from End-Member Mixing Analysis (EMMA). The main objective is to choose RR model that best complying with EMMA in terms of hydrograph separation that can be related to more accurate representation of real runoff generation processes.

**l.67-69: In the Introduction you criticized essentially all hydrological modles developed so far, yet you only test them on two very small catchments that are very close to each other, and are probably too small for a model relying on hydrological response units. So you cannot consider the performance for different climates, land use, geography, or size. What is the point of this study then, as related to the issues you raised in the Introduction?**

We criticize the different type of simplification in models which lead to different type of ambiguity in modelling results. In introduction, it was explained that we try to use an alternative approach for models' validation.

The appropriate HRU size is entirely subjective opinion but should be based on the data available for the object of study. In this particular case HRU were delineated according with data resolution.

The small homogeneous catchment guarantee the less noise contains in the useful signal (mainly geochemical) and the easier to make right conclusion from experiments. More area will blur measurements and complicate results interpretation. These advantages of the small test bed catchments are noted in the Introduction.

**l.77: You are at the same elevation as Hokkaido and you have very cold winters. Is it really tropical there?**

Tropical cyclones (typhoons) in the Northern Hemisphere can travel to high latitudes because of the presence of warm clockwise oceanic currents such as the Kuroshio.

[Figure]

https://earthobservatory.nasa.gov/images/7079/historic-tropical-cyclone-tracks
https://reliefweb.int/map/world/last-50-years-tropical-storms-asia-pacific-1966-2017

**l.84: Without explaining the symbols, equations are meaningless.**

We add an explanation for basic hydrological method Q=f(H), that is a water level-discharge rating curve, and for other cases.

**l.86: I checked the reference to find out about the fair profiles number but could not find an explanation. But I found an extensive modeling exercise with ECOMAG. To what extent does this paper repeat this reference?**

From the link https://link.springer.com/article/10.1134/S1064229321050057 you can check that in the study area of 45 km$^2$ there are 44 soil profiles, of which 14 are located directly on the studied catchments. In our opinion, this is quite a sufficient amount.

For these watersheds, the ECOMAG model was used for the first time.

**l.91: The suction you apply in such instruments determines which part of the pore space you sample, but you do not report this.**

**Suction cups cannot sample macropores unless these remain filled for a long period of time, which is typically not the case in unsaturated soils. Did your soils have macropores?**

**Did you remove these during the winter? (I am not sure they survive when they freeze.)**

Yes, the soil have macropores and due to this lysimeters were installed taking into account the local relief provide convergence of moisture flows and keep filling pore space as long as possible.

Observations on watersheds were carried out only in the warm season and lysimeters were removed for wintertime.

**l.96: ...Data quality control suggested next simulation periods... How exactly?**

Data with obvious errors associated with the operation of loggers were discarded.

**l.97, 98, 161, 181: unexplained abbreviations.**

We add explanation to all abbreviations even if they are widely known

**l.108-109: The end-member mixing analysis...hydrology methods... Unclear.**

End-Member Mixing Analysis (EMMA) is one of the well-known and widely used methods of tracer hydrology, the term often can be found in the literature

**l.110: ... hereafter called fractions... A fraction is very different from a source. You need to explain better what exactly you are doing.**

Terms will be checked and corrected.

**l.112: ...some empirical relations... Vague. And in the Introduction you stated that reliance on empirics was a weakness of current models.**

The use of empirical relationships and the use of empirical approaches in modeling are not the same thing. The remark concerns the fundamental points of the modeling methodology and reflects the misunderstanding due to the insufficiently clear presentation of the author's opinion in the text of the article. During the editing process, the authors will pay attention to a clearer wording of this position.

**l.123-124: ...water quality... I believe you mean the various substances dissolved in the water.**

You are right, thank you for this note.

**l.126-130: This explanation of the use of bivariate scatter plots needs to be explained much clearer. What property exactly are you plotting? Concentrations, fluxes, loads during a given period? And when you state all possible combinations need to be plotted a assume you mean all possible combinations of two, i.e., only solute pairs will be plotted.**

Preliminary bivariate scatter plots used for analyze the solutes. The note to be taken into account when editing the text.

**What will be the effect on colinearity if one solute is non-sorbing and the other is adsorbed? Both can still be conservative.**

The EMMA method implies the conservatism of tracers during mixing. The technique includes checking this condition on each specific data set.

**l.179: Please consult the guidelines for authors on the use of abbreviations in equations and the font of variables.**

Thank you for this note. Equations and abbreviations will be aligned with HESS compliance.

---

## Author Comment (AC2)

The authors express their deep gratitude to the referee for a thorough analysis of the article and a significant number of valuable comments. Criticism of the English language of the manuscript is fully accepted, the authors will make efforts to correct the text. The following are responses to individual comments.

**This manuscript details a study that compares runoff decomposition as estimated by end-member mixing analysis on one hand and hydrological models on the other hand. This issue is clearly of great interest for both hydrological processes understanding and model development. At this stage, the paper has several shortcomings in terms of methods and data, which prevents a full understanding of the paper. My suggestion would be to be less ambitious, e.g. in the number of hydrological models used but more exhaustive in the details given throughout the manuscript. Additionally, it should be noted that the English level is pretty poor, I am not a native English speaker but I recommend that the authors proofread their revised manuscript before submission.**

The main idea of the article was precisely that several widely used runoff models are compared, which are parameterized by conventional, standard methods, completely independently of one to another and something else. Initially the authors decided not to pay too much attention to the standard descriptions, it to be done. Then, from the resulting simulated series, data on runoff components are extracted, which are direct depending on the model used. These data are compared with runoff components obtained from EMMA, on the basis of which the authors try to draw conclusions about the greater or lesser adequacy of different models. The question that the article tries to answer is the following: is such an approach promising in principle? Thus, the comparison of just a set of models seems to be the key for the author's approach.

**Major comments**

**Lack of details**

**Throughout the paper, there is a lack of details. This affects both the data/method section and the results/discussion section at a level that precludes the reader to provide clear guidelines for further improvements. The required additional information is listed in the minor comments hereafter. The two other major comments are related to methodological issues.**

The comment is accepted, the text will be improved.

**Hydrological model uncertainties and how the methodology of the paper reduces it**

**As explained in the introduction (l.31-39), hydrograph decomposition may be a powerful tool to reduce equifinality. In this sense, the present study shows relatively similar runoff simulations but with different flow components from hydrological models, but with different flow components. Unfortunately, the authors did not take this opportunity seriously, they used a single optimal parameter set for each model and did not discuss the impact of this choice, nor the way the parameter set is optimized. Consequently, it is pretty hard to conclude the relative weights of structural and parameter uncertainties in explaining the results.**

The possibilities of the authors' analysis are limited by the volume of available information. For each case, only three years were used when comparing data for both runoff modeling and EMMA analysis. Although there is a little more data only for calibration, it is still not enough for various fine measurement methods, comparison of parameter sets, etc. e. This shortcoming the authors try to compensate by using several models and taking into account the composition of the runoff.

**Short record periods**

**Only three years are available for model simulation (what about the warm-up year?). It is quite short and consequently, no validation was performed by the authors. Modeling results are presented only for calibration, which is problematic when dealing with parameter/structural uncertainties. Also, as low-flow components are extracted from hydrological models simulations, the authors should verify cautionly model initialization.**

The answer to the question is partially given above. An analysis of the literature shows that data sets that allow continuous daily runoff decomposition over several years are extremely rare. Thus, the noted lack of data is an objective circumstance and should be overcome in the course of further work.

Four years including warm-up for each catchments were used. One year for warm-up definitely should be enough for properly initialize small catchment models.

**Minor (but still important) comments**

**l.77-78: not clear what is the time step of heavy rainfall and how extreme are these events.**

In the description of the objects of study, we indicate:
l.73: "The annual average precipitation amount is 700-800 mm"
l.78: "The range of maximal daily heavy rains is 100–200 mm."
Thus, by comparing the average annual precipitation and for an individual rain, one can draw a conclusion about the extremeness of such events.

**l.79: not clear how averaging is performed, spatial or temporal?**

It means sum (temporal) of precipitation for rain, measured by rain-gauge.

**Please add a table with both catchments characteristics (mean annual rainfall, temperature, runoff, land use lithology, topography, etc.). The differences in runoff yields for these two neighbor catchments are huge and I cannot figure out if it is due to lithological differences or specificity of the (short) record periods with extreme events.**

This notes to be taken into account when editing.
We present the table 1 following with information about catchments.
The differences are rather due to the geological structure than a short series of observations

**Table 1**. Catchment's characteristics.

| Characteristics | Elovy | Medvezhy |
|---|---|---|
| Area, km2 | 3.5 | 7.6 |
| Avg. Height, m | 722 | 704 |
| Max Height, m | 962 | 869 |
| Avg. slope, % | 13.5 | 13.8 |
| Max. slope, % | 28.7 | 31.5 |
| Avg. Precipitation, mm/day * | 2.13 | 2.35 |
| Avg. Temperature, C * | 3 | 3.23 |
| Avg. Discharge, mm/day * | 0.65 | 0.75 |
| Land-use | Coniferous-broadleved and coniferous forest | Coniferous-broadleved forest |
| Lihology | Cretaceous volcanites (tuffs) and sub-volcanic acid and intermediate rocks (granites, ryolite, porphyrites and diorites) | Jurassic metamorphic basic rocks (gabbroids etc.) |

*daily values, assessment period: 01.01.11-31.12.14 for Elovy creek and 01.01.14-31.12.17 for Medvezhy creek

**Figure 1. Where the WMO station is located?**

This will be taken into account when editing. Since the weather station is located 30 kilometers away, we add the coordinates: WMO 31939 (Chuguevka) weather station located in 35 km to the NW from the observation sites (E 133°53'48" N 44°11'59").

**l.132: It is not clear how the end members are identified, what is the "independent information"?**

In this case, it's a lapse. This meant (but wrongly expressed) the data on potential sources that were not included in the series of river water data. It will be edited.

**Please add a table with the characteristics of the three hydrological models (with e.g. basis of the snow components, number of free parameters, spatial and temporal discretization, etc.). The information given for each model is not homogenous. Nothing is said on parameter estimation, which is in my opinion a key issue (see major comment #2).**

The authors will make an effort to present homogeneous information on the three applied hydrological models (Table 2).

Table 2. Hydrological models characteristics

| Characteristics / Model | ECOMAG | SWAT | HBV |
|---|---|---|---|
| Spatial discretization | HRU | HRU | Lumped |
| Temporal discretization | Daily | Daily | Daily |
| Number of calibrated parameters | 9 | 12 | 10 |
| Snow component | Degree-day | Degree-day | Degree-day |
| Evaporation | Dalton method | Penman-Monteith | Penman-Monteith |
| Surface flow | Kinematic wave | Kinematic wave | Storage based |
| Soil flow | Darcy's law | kinematic storage model | Storage based |
| Groundwater flow | Darcy's law | Storage based | Storage based |
| Routing | Kinematic wave | Variable travel time | Triangular weighted |

**l.238-249: Are the results are shown in calibration (and by the way, how the calibration is performed)?? Please modify Figure 5 so that the reader can see the whole record period simulation results.**

The remarks above relate to the design of materials and the detail of the presentation of the article. In general, comments are accepted, the text of the article will be improved taking into account these shortcomings.

Fig. 5 shown results of calibration. Calibration was performed manually for all models. Figure 5 was modified to represent all simulation period (without warm-up year).

**Table 4: it appears that the models present quite different flow decompositions. Could this be due to the fact that the a priori three-component is wrong because too detailed for such small catchments? Since many flow decompositions only concern two flow components ("baseflow" and "surface flow"), did the authors challenge their prior 3-components hypothesis?**

The hypothesis of three main runoff components (conditionally "direct flow", "soil flow" and "ground flow") is generally accepted and widely confirmed in tracer studies based on EMMA. The text of the article indicates that the EMMA analysis for the studied basins quite clearly showed three runoff

components in one case and four in the other. As for simulation models, the number of runoff components in them is determined by the structure of the model. Only 3-component runoff models were specially selected for comparability of simulation results with EMMA decomposition. The variety of runoff composition indicated by the reviewer, obtained in various models, is precisely the key problem considered in the article. For larger basins, this diversity is no less than for the smallest ones. The indicated problems are also strongly conditioned by the lack of unification of theoretical ideas about the runoff and the corresponding terminology, which is discussed in sufficient detail in the text of the article. This lack cannot be compensated within the scope of this article and must be taken into account in the evaluation of the results.

**l.359-360: These perspectives are quite fuzzy. Please provide a real discussion section in the paper. There is a lot to say on both methodological limitations and further analysis of the results obtained.**

The Discussion section will be checked and edited.